# Clarification of 1,3-Propanediol Fermentation Broths by Using a Ceramic Fine UF Membrane

**DOI:** 10.3390/membranes10110319

**Published:** 2020-10-30

**Authors:** Wirginia Tomczak, Marek Gryta

**Affiliations:** 1Faculty of Chemical Technology and Engineering, West Pomeranian University of Technology in Szczecin, ul. Pułaskiego 10, 70-322 Szczecin, Poland; 2CEA, DEN/DEC, 13108 Saint-Paul-lez-Durance, France

**Keywords:** 1,3-propanediol, post-fermentation solution, resistance-in-series, titanium membrane, ultrafiltration

## Abstract

This work examined the use of a ceramic fine ultrafiltration (UF) membrane for the pre-treatment of 1,3-propanodiol (1,3-PD) fermentation broths. It has been demonstrated that the membrane used provides obtaining a high-quality, sterile permeate, which can be sequentially separated by other processes such as nanofiltration (NF) and membrane distillation (MD). Special attention was paid to the impact of the operational parameters on the membrane performance. The series of UF experiments under transmembrane pressure (TMP) from 0.1 to 0.4 MPa and feed flow rate (Q) from 200 to 400 dm^3^/h were performed. Moreover, the impact of the feed pH, in the range from 5 to 10, on the flux was investigated. It has been demonstrated that for fine UF, increasing the TMP is beneficial, and TMP equal to 0.4 MPa and Q of 400 dm^3^/h ensure the highest flux and its long-term stability. It has been shown that in terms of process efficiency, the most favorable pH of the broths is equal to 9.4. An effective and simple method of membrane cleaning was presented. Finally, the resistance-in-series model was applied to describe resistances that cause flux decline. Results obtained in this study can assist in improving the cost-effectiveness of the UF process of 1,3-PD fermentation broths.

## 1. Introduction

1,3-Propanediol (1,3-PD; CH_2_(CH_2_OH)_2_), an organic compound, has gained considerable attention owing to its myriad industrial applications. Indeed, 1,3-PD is widely used as a monomer for the chemical synthesis of polyethers, polyurethanes and polyesters [1,2,3,4,5,6,7]. In addition, it is applicable in the production of different materials, such as composites, detergents, lubricants, laminates and coatings [8,9]. Besides, it is used in many other fields, for instance pharmaceutical, food and textile industries [5,10]. It is worth noting that the production of 1,3-PD has been growing rapidly and it has been estimated that in 2022, its global market will grow from 490 million USD in 2019 to 870 million USD by 2024 [11].

Since chemical synthesis of 1,3-PD is expensive and environmentally unfriendly [1,6,8,12,13,14], many efforts have been made to investigate its production through microbiological bioconversion, which allows the use of renewable feedstock and provides no generation of toxic by-products. Glycerol, generated as a major by-product during biodiesel production, is considered the best natural substrate that can produce 1,3-PD [15]. Indeed, Crosse et al. [16], in their recently published review article, have pointed out that almost 20% of all scientific papers pertaining to crude glycerol mention 1,3-PD. It has been well-documented that among the bacterial strains that can convert glycerol into 1,3-PD via hydrogenolysis possesses are: *Citrobacter* [3,17,18,19,20], *Klebsiella* [3,12,21,22], *Clostridia* [1,13,23,24], *Lactobacilli* [8,9,25,26,27] and *Enterobacter* [28,29].

However, the development of a commercially viable process to produce 1,3-PD by fermentation requires an efficient and economically useful technique that would allow its recovery from fermentation broths. Hence, separation of post-fermentation solutions is an important challenge that involves several integrated processes [30,31]. It is due to the fact that fermentation broths are very complex media characterized by variety and content of many different components. Indeed, in addition to the main product, 1,3-PD fermentation broths contain: water, residual substrate (e.g., glycerol), by-products (e.g., ethanol, 2,3-butandiol, acetate, lactate and succinate), macromolecules (e.g., protein, polysaccharide, nucleic acid) and salts [32,33]. The difficulties in recovery of 1,3-PD from complex media can also be attributed to its very hydrophilic nature, low volatility and a high boiling point, equal to 214 °C at atmospheric pressure [14,33,34]. Therefore, as it has been described in References [32,35], the downstream processing of fermentation broths to obtain high overall purity of 1,3-PD requires three main steps. The first one involves the removal of microbial cells, which can be performed by the membrane filtration. Then, the removal of impurities and primary separation of 1,3-PD from the broth is conduced. For this purpose, nanofiltration (NF) and membrane distillation (MD) processes can be applied [36]. The final purification of 1,3-PD can be achieved by vacuum distillation [37] and silica gel chromatography [38,39].

Importantly, the quality of the permeate obtained in the first step determines the effectiveness and efficiency of the above-mentioned separation processes. High-quality and sterile permeate, which can be further processed, can be obtained by using the ultrafiltration (UF), which is a pressure-driven membrane technique providing the separation of dissolved and suspended species based on the size and molecular scale. As a result, macromolecules with nominal molecular diameters higher than molecular weight cut-off (MWCO) of the membrane are concentrated in the retentate, while smaller molecules pass the membrane pores freely and are released into the permeate. Generally, UF membranes have pore size in the range from 2 to 100 nm which retain molecules with molecular weights from 350,000 to 1000 Da [40]. 

Performing a comprehensive review related to the UF of fermentation broths by using polymeric and ceramic membranes (Appendix A, Table A1) showed that UF is an excellent technique for purification of post-fermentation solutions with various main products, such as: surfactin [41,42,43,44,45], clavulanic acid [46], prodigiosin [47], lactic acid [48,49,50], succinic acid [51], demethylchlortetracycline [52], hyaluronic acid [53] and xanthan [54]. It is worth noting that in Reference [51], the UF process was apply to clarify the succinic acid fermentation broths as the first operation unit in a succinic acid bio-refinery. For this purpose, the authors have used three polyethersulfone membranes with MWCO equal to 10, 30 and 100 kDa, as well as one regenerated cellulose membrane with MWCO of 10 kDa. Importantly, it has been shown that UF ensures obtaining a clearer permeate compared with that from centrifugation. Indeed, it has been reported that although microorganism cells were removed in both UF and centrifugation, the protein removal rate of ultrafiltration (86%) was higher than that of centrifugation (53%). Moreover, the authors have noted that membrane material has a more significant impact on the permeate flux than membrane MWCO. Although all membranes tested provided good clarification performance, the regenerated cellulose membrane has been indicated as the most appropriate for the studied fermentation broths’ clarification. It is due to the fact that this hydrophilic membrane showed the highest membrane flux. In turn, Mubarak et al. [44] have investigated a recovery and purification of surfactin from the fermentation broths with *Bacillus subtilis* ATCC 21332 by a single-step cross-flow UF with using a hydrosart and polyethersulfone membranes with MWCO equal to 10 kDa. The broths were composed of: biomass (2.06 ± 0.02 g/dm^3^), surfactin (447.06 ± 1.25 mg/dm^3^) and protein (109.00 ± 2.11 mg/dm^3^). The authors have shown that both membranes used led to the high recovery and purity of surfactin. It should be pointed out that the results obtained in the above-mentioned study can allow to reduce the overall cost of surfactin production. In turn, Li et al. [49] have investigated the application of the cross-flow UF system for separation of cells and proteins from cheese whey broths containing: lactose (2.2 g/dm^3^), lactic acid (25.6 g/dm^3^) and acetic acid (1.0 g/dm^3^). It has been indicated that increasing the MWCO of the polyethersulfone membrane from 5 to 20 kDa led to a significantly higher permeate flux and lower crude protein retention ratio. In addition, the authors have shown that the effect of membrane MWCO on the protein retention ratio is not significant. 

It must be recognized that the presented literature review (Appendix A, Table A1) shows that the investigations of the purification of 1,3-PD fermentation broths by UF process are limited only to one study [55], where a ceramic membrane with MWCO equal to 8 kDa was used. Moreover, regarding the UF process of fermentation broths with other main products, most of the information comes from experiments performed by the use of the following polymeric membranes: polyethersulfone [41,42,43,44,47,48,49,56,57], cellulose [41,42,44,45,47,48,51], polysulfone [54,58], fluoro polymer [52] and polyvinylidene fluoride [52,53], while the open literature contains only two studies [46,50] investigating the use of ceramic membranes. 

The conducted critical review of the published articles leads to the conclusion that there is a need to extend the research into the use of ceramic ultrafiltration membranes for the purification of 1,3-propanediol fermentation broths. The motivation lies in the fact that the downstream processing in biotechnological processes may constitute a major part of the total production cost [59]. One of the factors that significantly determine the profitability of the pressure-driven membrane processes is the value of the obtained permeate flux. It has to be pointed out that membrane fouling is a great concern for application of membrane technology [60,61,62]. However, it is well known that this inevitable phenomenon can be controlled and alleviated by determining the most favorable values of operational parameters, such as transmembrane pressure (TMP) and feed flow rate (Q). 

The use of ceramic membranes in purification of fermentation broths is of great interest because of their many advantages over the polymeric membranes. It is well documented [63,64,65,66,67,68] that ceramic membranes offer higher permeability values and reduced tendency to fouling, which is due to their relatively narrow pore size distribution as well as higher porosity and hydrophilicity nature. Moreover, ceramic membranes demonstrate high bacterial resistance [63,64]. Also, they hold excellent acid and alkali tolerance [69,70], thus they can be regenerated by various chemical products. In addition, as described in References [68,71,72,73,74,75,76,77,78], ceramic membranes have mechanical and thermal resistance which allows to perform filtration processes under severe operational conditions (high transmembrane pressure, feed flow rate and temperature) as well as enables an aggressive membrane cleaning. Finally, although generally ceramic membranes are more expensive than polymer membranes [63], they show longer service life with much reduced maintenance [79]. The above-mentioned advantages of ceramic membranes indicate their potential in the industrial purification of post-fermentation solutions.

The present study is part of the development process of 1,3-propanediol production through fermentation of glycerol. The research shows the application of the ceramic fine ultrafiltration membrane for purification of 1,3-propanediol post-fermentation solutions from *Citrobacter freundii* cells and macromolecular substances. The work aims to find a set of operational conditions for the transmembrane pressure (TMP) and feed flow rate (Q) that ensure the highest permeate flux rate and long-term performance stability. Moreover, the impact of feed pH on the permeate flux has been studied. In addition, an effective, short and simple method of membrane cleaning was presented. Finally, to analyze resistances that lead to the flux decline during the performed UF runs, the resistance-in-series model was applied.

## 2. Materials and Methods

### 2.1. Fermentation

The microorganism used to produce 1,3-propanediol was a strain of *Citrobacter freundii*, which was isolated and characterized in the Department of Biotechnology and Food Microbiology, Poznań University of Life Science, Poland. The bacteria were inoculated under sterile conditions in a bioreactor (5% vv.) and grown on medium containing, per liter: 20 g glycerol as a carbon source, 3.4 g K_2_HPO_4_, 2.5 g peptone K, 1.5 g meat extract, 2.0 g (NH_4_)_2_SO_4_, 2.0 g yeast extract, 1.3 g KH_2_PO_4_, 0.4 g MgSO_4_·7H_2_O, 0.08 g CaCl_2_ and 0.002 g CoCl_2_. The fermentation was conducted during two days under temperature equal to 30 °C and agitation at 150 ± 5 rpm. The initial pH of the medium was equal to 7. To evaluate the impact of the feed pH on the permeate flux, the pH of the broths was adjusted to the desired pH values before each experiment, by adding a portion of NaOH solution (1 mol/dm^3^).

### 2.2. Ultrafiltration Unit

The experiments were performed in a continuous mode of operation in a pilot-scale apparatus (INTERMASZ, Września, Poland) equipped with a cross-flow UF membrane module (Figure 1). The scheme of the pilot-scale installation was presented and described in our recent study [80].

The purification process was done using the single channel tubular ceramic fine UF membrane manufactured by TAMI Industries (Lyon, France). The membrane selective layer was titanium deposited over an alumina support. It had a nominal cut-off equal to 450 Da. It was 21.5 cm long with an external and an internal diameter of 10 and 7 mm, respectively. The membrane surface area was equal to 0.0047 m^2^. The detailed specification of the used membrane is presented Table 1.

### 2.3. Operating Conditions and Procedure of Membrane Cleaning

Ultrafiltration was evaluated as a purification process for fermentation broths containing 1,3-propanediol as a main product. The experiments have been performed under constant transmembrane pressure, defined according to the following equation:(1)$$\mathrm{TMP}=\frac{P_{\mathrm{IN}}+P_{\mathrm{OUT}}}{2_{}}-P_{P}$$
where P_IN_, P_OUT_ and P_P_ are a pressure in the inlet, outlet and on the filtrate membrane side, respectively.

Before every test, the pure water flux of the clean membrane in function of transmembrane pressure at temperature 30 °C was measured. In order to investigate the impact of process parameters on the membrane performance, the series of UF experiments under TMP in the range from 0.1 to 0.4 MPa and Q from 200 to 400 dm^3^/h were performed. The applied values of the feed flow rate correspond to the flow velocity, u, from 1.44 to 2.88 m/s and Reynolds number (Re) from 11,859 to 23,718. Temperature was kept constant at 30 °C by a heat exchanger connected to tap water. Each UF experiment lasted 150 min, time enough to reach steady-state permeate flux.

The rate of permeation through the membrane was characterized as the permeate flux calculated according to the formula:(2)$$J=\frac{1}{S}\frac{\mathrm{dV}}{{\mathrm{dt}}_{}}$$
where: J is permeate flux (dm^3^/m^2^h), V is volume (dm^3^), S is area of active membrane (m^2^) and t is time (h).

The turbidity rejection (R) was calculated as follows:(3)$$R=(1-\frac{\tau_{P}}{\tau_{F}})\times 100\%$$
where *τ*_P_ and *τ*_F_ are the turbidity of permeate and feed samples, respectively.

Since flux decline during each UF experiment was observed, between different filtration runs, the fouled membrane was cleaned in multiple steps to recover its initial permeate flux. Immediately after each UF process, the fermentation broth was drained from the system. Then, the system was rinsed with distilled water to remove any residual solution and labile surface deposit. Afterwards, chemical cleaning was performed. In the current study, chemical cleaning by using 1% w/w solution of NaOH has been performed. Although there are many types of agent available for ceramic membrane cleaning, in the present study, caustic solution has been used since it is typically used to clean membranes fouled by organic and microbial solutions [81]. After the chemical cleaning, the system was recycled with distilled water to rinse away the residual detergent. Each of the steps mentioned above were carried out for 5 min at 30 °C, under the feed flow rate equal to 200 dm^3^/h and zero transmembrane pressure. It is important to note that the permeate outlet was closed (TMP = 0) since it prevents the adsorption, penetration or compact of the pollutants in the membrane pores [82]. After the cleaning, permeability of the membrane was completely restored, and it was used again for the next tests. It demonstrates that the proposed method of membrane cleaning is effective, short and simple, which may have a positive impact on the economics of the UF process of 1,3-PD fermentation broths.

### 2.4. Analytical Methods

The samples of the fermentation broths and obtained permeate were analyzed for the content of compounds (1,3-propanediol, glycerol and by-products), pH, turbidity, viscosity and the number of bacteria. The analytical methods have been described in detail in our previous study [80].

### 2.5. Resistance-in-Series Analysis

In the recently published review paper, Di Bella and Di Trapani [83] have pointed out that the resistance-in-series model is likely the most complete and applied model to perform fouling investigations. Therefore, in the present study, this model has been used to identify the contribution of different types of fouling resistance during the UF process of 1,3-PD fermentation broths with *Citrobacter freundii*. In general, permeate flux through the fouled membrane can be described in terms of the total resistance, R_T_, as follows:(4)$$J=\frac{\mathrm{TMP}}{\eta_{P}\cdot R_{T}}$$
where *η*_p_ is permeate viscosity.

The total resistance is computed as the sum of three different resistances, as shown in the following formula:R_T_ = R_m_ + R_irr_ + R_rev_(5)
where R_m_, R_iir_ and R_rev_ are the resistance of clean membrane, irreversible and reversible fouling, respectively.

R_m_ is constant, whereas the R_iir_ and R_rev_ are dependent on the feed quality and the membrane operating conditions [84]. Roughly speaking, R_irr_ is related to pore blocking and its removal requires a chemical cleaning of the membrane, while R_rev_ is caused by the concentration polarization and accumulation of the cake layer on the membrane surface, and it is removable by water rinsing [85,86,87]. The parameters in Equation (5) have been determined based on the description presented in References [55,85,88,89,90] (Table 2), where membrane flux decline has been evaluated during the low-pressure membrane techniques used for purification of various biological suspensions.

## 3. Results

### 3.1. Physico-Chemical Characterzization of the Post-Fermentation Solutions

The results of the analysis of the broth media composition, obtained with the use of liquid and ion chromatographs, are presented in Table 3 and Table 4. It has been determined that fermentation broths contained 1,3-PD and the following by-products: citric acid, lactic acid and acetic acid. Moreover, the presence of the ions: Cl^−^, NO_3_^−^, PO_4_^3−^, SO_4_^2−^, Na^+^, NH_4_^+^, K^+^, Ca^2+^ and Mg^2+^ has been noted. Since compounds dissolved in the feed were not rejected by the membrane, the solute concentrations in the feed and permeate were the same.

Initial pH of the medium was equal to 7. However, after two days, as a result of organic acids formation, the analyzed broths were characterized by an acid pH, in the range between 4.5 and 5.5. The noted turbidity was in the range from 145 to 230 NTU. The dynamic viscosity has been reported as 0.85 × 10^−3^ Pa·s. In turn, number of bacteria was in the range between 9.0 and 10.5 log CFU/mL and the total wet biomass was determined in the range from 7.02 to 9.98 g/dm^3^.

### 3.2. Purification Efficiency

The bibliographic review of the state-of-the-art, which was briefly discussed in the first section, indicated that the quality of the permeate obtained in the pre-treatment operation determines the effectiveness and efficiency of subsequent processes of fermentation broths’ separation, such as NF and MD. Therefore, obtaining a very high-quality permeate was one of the main goals of the performed UF experiments.

The fermentation broths contained a large number of microorganism cells and metabolites. The efficiency of the bacteria cells removal from the fermentation broths has been defined by the bacteria count in the obtained permeate. It was controlled under all applied process conditions and for various feed quality. Generally, UF membranes are known to be efficient for the elimination of bacteria. It is due to the difference in size between the bacterial cells and the membranes’ pore. This fact has been proved in the present study, since, as expected, no bacteria cells have been detected in the permeate samples. It clearly demonstrates that the use of ceramic fine ultrafiltration membrane with the MWCO equal to 450 Da provides for obtaining the sterile permeate.

In turn, the efficiency of removal of macromolecular substances was estimated based on measurements of the feed and permeate turbidity. Figure 2 shows the reported values of the feed turbidity and total wet biomass (a) and permeate turbidity (b) during the run of UF process performed under TMP and Q equal to 0.2 MPa and 300 dm^3^/h, respectively. At the beginning of the process, the feed turbidity was equal to 130 NTU. Along with the duration of the purification process, it increased, and in the end of the process, it was equal to 625 NTU. The observed increase can be justified by the fact that the UF process led to the broth thickening. Indeed, the total wet biomass during the investigated process increased from 10.01 to 27 mg/dm^3^. In contrast, the turbidity of permeate decreased significantly with the experiment run. It has been found that at the beginning of the process, it was equal to 0.9 NTU, and after 60 min, it was reduced twice. In turn, in the end of the process, it was lower than 0.1 NTU. It is related to the fact that with increasing the feed turbidity, the cake layer formed by the biomass and other broth components was thicker and acted as a second membrane [42]. Thus, it improved the permeate quality through a higher efficiency in removing macromolecular substances. This phenomenon has also been observed in our recently published study [80], where the cross-flow microfiltration (MF) of 1,3-PD fermentation broths has been investigated.

Results obtained in this study showed that the UF membrane achieved high turbidity rejection, equal to 99.98%. Therefore, it clearly indicates that the membrane used is characterized by the excellent separation properties allowing to retain all macromolecular substances present in the studied fermentation broths. It has to be pointed out that although in the present study many UF tests have been carried out, under different process conditions and for a feed of different quality, at the end of each UF run, the same value of the permeate turbidity has been recorded. It clearly demonstrates that obtaining a high-quality permeate is independent of the feed turbidity and it is ensured under all applied operational conditions. Therefore, it can be concluded that both feed quality and operational parameters do not affect the membrane selectivity.

It is worth noting that the quality of the permeate obtained in the present work is higher than that noted in the previous studies where purification of 1,3-PD fermentation broths have been performed by using MF [80] and UF [55] membranes. In Reference [80], it has been shown that microfiltration membrane with the nominal pore size equal to 0.14 um allows obtaining permeate characterized by the turbidity equal to 0.20 NTU. In turn, in Reference [55], it has been found that the use of an ultrafiltration membrane with the molecular weight cut-off equal to 8 kDa ensured to obtain the permeate with the turbidity of 0.1 NTU. The conducted comparison of the permeate turbidity obtained in the current research and our previous studies confirms the well-known fact that the quality of the permeate is closely related to the pore size of the membranes used for the purification processes.

Results obtained in this study clearly show that the membrane used provides obtaining the high-quality permeate, which can be sequentially separated by NF and MD processes. Therefore, it can be concluded that fine UF membranes with the MWCO equal to 450 Da have the industrial potential for the pre-treatment of 1,3-propanediol fermentation broths from *Citrobacter freundii* bacteria and macromolecular substances.

### 3.3. Impact of Transmembrane Pressure and Feed Flow Rate on the Permeate Flux

It is well known that decreasing of permeate flux during membrane processes is the most important limitation for membrane technology. Indeed, the authors of the recently published comprehensive statistical review [91] have indicated that fouling of UF membranes has been an area of particular interest in the last 10 years. Indeed, this issue has been reported in several studies [41,42,43,46,47,48,49,50,51,52,53,54,55,56,57] where the UF process of fermentation broths with various microorganisms has been investigated. As an example, Wang et al. [51] have demonstrated the significant reduction of UF efficiency during the clarification of succinic acid fermentation broth with *Actinobacillus succinogenes*. The authors have reported that during the process performed under 0.2 MPa, the permeate flux of PES membrane with MWCO of 30 kDa decreased from 65.2 to 14.65 dm^3^/m^2^h.

Therefore, and unsurprisingly, in the present study, a significant reduction of the membrane performance during UF experiments has been found. Figure 3 shows changes of the permeate flux during the filtration process conducted under TMP equal to 0.1 MPa and Q of 200 dm^3^/h. In general, three phases of flux decline have been identified. The first phase (up to 30 min of the process) was characterized by the most notable permeate flux decline, from its maximum value equal to 47.62 to 10 dm^3^/m^2^h. In the second phase (from 30 to 50 min), the flux continues to decline, however, less intensively. During the period mentioned, the permeate flux was recorded to decrease from 10 to 4.25 dm^3^/m^2^h. In turn, in the third phase, a quasi-steady-state period, the permeate flux was considered as stabilized, and by the end of the process, it was equal to 4.25 dm^3^/m^2^h. These results indicate that the obtained value of the permeate flux constitutes only 9% of its initial value. The key highlight is therefore that performing the UF process of post-fermentation solutions with *Citrobacter freundii* requires determination of the conditional parameters, which ensure obtaining the highest membrane performance. In particular, selecting the suitable TMP and Q is fundamental for guaranteeing that the membrane module operates under the most favorable conditions. Thus, another aim of the present study was to determine the impact of conditional parameters, such as TMP and Q, on the membrane performance. For this purpose, several experiments, under various TMP and Q values respectively, in the range from 0.1 to 0.4 MPa and from 200 to 400 dm^3^/h, have been conducted.

Figure 4 shows the impact of TMP and Q on the steady-state permeate flux during the UF process. For instance, increasing the TMP from 0.1 to 0.2 MPa led to an increase in the steady-state permeate flux from 4.25 to 6.34 dm^3^/m^2^h under the feed flow rate equal to 200 dm^3^/h. A further increase in TMP to 0.3 and 0.4 MPa allowed increasing membrane performance to 7.70 and 8.55 dm^3^/m^2^h, respectively. In turn, performing the UF process under 400 dm^3^/h and TMP of 0.1 MPa allowed to obtain the steady-state permeate flux equal to 17.79 dm^3^/h, whereas under the TMP of 0.4 MPa, the membrane was characterized by the performance of 31.78 dm^3^/m^2^h. These results indicate that TMP affects the performance of the membrane used in the presented experiments. Permeation fluxes are clearly higher for the higher TMP, which is supported by the greater hydrodynamic driving force toward the membrane surface. The results obtained in the present study are in agreement with those presented in References [43,47,49,54,55,56], where increase of the permeate flux with increasing of TMP during UF of various fermentation broths has been reported. However, it is worth noting that in the present study, the membrane flux increased with increasing TMP in a near-linear relationship until a limiting flux value was reached. Then, the membrane flux increased slightly. It has been determined that the value of the critical flux depends on the feed flow rate. It was equal to 4.25, 9.43 and 17.79 dm^3^/m^2^h under Q of 200, 300 and 400 dm^3^/h, respectively. Since the obtained relation between permeate flux and TMP is below that reported for the pure water, limiting fluxes noted during all performed experiments are classified as a weak form of the critical flux [92]. This type of critical flux has also been reported in previous studies [42,43,47,55,93], where low-pressure membrane processes (MF and UF) have been used to perform the pre-treatment of fermentation broths.

Moreover, it has been demonstrated that the feed flow rate can also have an important effect on the permeate flux during UF of fermentation broths. It is observed from Figure 4 that, for instance, increasing of the feed flow rate from 200 to 300 dm^3^/h at TMP equal to 0.1 MPa led to a slight increase in the steady-state permeate flux (from 4.25 to 5.07 dm^3^/m^2^h). However, a further increase of the feed flow rate to 400 dm^3^/h resulted in a 4-fold increase in the flux (to 17.79 dm^3^/m^2^h). In turn, under the +highest TMP applied (0.4 MPa), for the flow rate of 200, 300 and 400 dm^3^/h, the obtained steady-state permeate flux was equal to 8.55, 14.12 and 31.78 dm^3^/m^2^h, respectively. These results clearly indicate that selecting a suitable feed flow rate can control membrane fouling and thus improve membrane performance. Demonstrated improvement in permeate flux with the feed flow rate increment is due to the increase in shear force near the membrane. It can lead to sweeping away the accumulated solutes and returning them back to the feed solution. This, consequently, results in a reduction in the mass and thickness of the cake layers formed on the membrane surface. Owing to this fact, increasing in Q may result in a reduction of filtration resistance [94]. These results are consistent with the findings of Marzban et al. [57], who showed that the permeate flux increases with an increase in the feed flow rate during micro- and ultra-filtration processes of *Bacillus thuringiensis* fermentation broths performed under constant pressure. However, it should be pointed out that in the present study, under all applied TMP, the most significant increase of membrane performance has been noted for increasing of the feed flow rate to 400 dm^3^/h. It suggests that the flow rates below 400 dm^3^/h did not effectively weaken the fouling layer deposited on the membrane surface. It can be explained by the fact that the feed flow rate equal to 200 and 300 dm^3^/h provided a flow regime that is applicable to laminar (Re equal to 11,859 and 17,788, respectively), whereas the feed flow rate of 400 dm^3^/h corresponded to the transition regime (Re equal to 23,718).

Conducting series of UF experiments of 1,3-PD fermentation broths under a wide range of operational conditions allows to indicate that performing the UF process under TMP equal to 0.4 MPa and Q of 400 dm^3^/h provides the highest permeate flux, equal to 31.78 dm^3^/m^2^h.

### 3.4. Impact of Feed pH on the Permeate Flux

It is well known that pH of the solution is a significant factor affecting ceramic membranes’ performance. It is due to the fact that when a metal oxide is in contact with an aqueous medium, membrane surface groups can undergo the process of dissociation, resulting in changing of the membrane surface charge and thus in foulant–membrane interactions [95,96,97,98]. Many efforts have been made to investigate the impact of feed pH on the performance of ceramic membranes used for ultrafiltration of various solutions, such as textile wastewater [71], municipal wastewater [72], oil-in-water emulsion [74], yeast suspensions [75], proteins and polysaccharides [97], seawater [99,100], model solutions of humic acid [97,101] and pineapple’s crude waste mixture [102]. Based on the aforementioned examples above, it can be inferred that pH of fermentation broths also has a significant impact on the fine UF membranes permeate flux. While of critical interest, to the best of our knowledge, there are no reports showing the influence of pH of fermentation broths on the ceramic UF membranes performance. Consequently, the importance of this study is related to the determination of the impact of the pH of 1,3-PD broths with *Citrobacter freundii* on the fine UF membrane permeability. In order to reveal how the permeate flux depends on solution pH, the UF experiments have been conducted for broths characterized by six different pH values equal to 5, 6, 7, 8.5, 9.4 and 10. The processes have been performed under TMP equal to 0.4 MPa and Q of 400 dm^3^/h, since, as it has been demonstrated in Section 3.3., the above-mentioned conditions ensure the highest permeate flux.

It can be clearly observed from Figure 5 that, generally, pH of the fermentation broths affects the permeability of the membrane used in the present study. It has to be pointed out that the increase of pH from 5 to 6 did not lead to the change in membrane performance. On the other hand, it has been noted that increasing the pH from 6 to 9.4 resulted in an almost 2-fold increase in the permeate flux, from 31.78 to 62.50 dm^3^/m^2^h. It indicates that the isoelectric point (EIP) of the membrane is located at pH of 6. Hence, the membrane surface charge is positive at the pH range lower than this value, while it becomes negative in the higher pH. For the membrane used in the present study, with an active layer made up of amphoteric material titanium oxide, it can be expressed as follows [103]:(6)$$-\mathrm{Ti}-\mathrm{OH}+H_{3}O^{+}↔-\mathrm{Ti}-{\mathrm{OH}}_{2}^{+}+H_{2}O\;\mathrm{pH}\;<\;6$$
(7)$$-\mathrm{Ti}-\mathrm{OH}+{\mathrm{OH}}^{-}↔-\mathrm{Ti}-O^{-}+H_{2}O\;\mathrm{pH}\;>\;6$$

Consequently, for the feed pH values higher than 6, the interactions between components of fermentation broths and the membrane decreased, which led to a reduction in the intensity of cake deposition onto the membrane surface and thus an increase in final steady-state permeate flux. On the contrary, positively charged promotes foulants adsorption on the membrane surface and/or inside its pores. These findings are in agreement with results presented in References [76,103,104], where authors have shown that the IEP of titanium oxide membranes is located at pH equal to around 6.

It can be concluded that the adjustment of the feed pH can cause a change in the interactions between feed and membrane, leading to a significant enhancement of ceramic membrane performance during the UF process of 1,3-PD fermentation broths. In the current study, the most favorable feed pH allowing to obtain the highest process efficiency under the above-mentioned conditions is equal to 9.4. Further increasing the feed pH did not lead to an increase of the membrane performance.

### 3.5. Resistance Analysis

Membrane resistance, R_m_, was calculated from the water flux and transmembrane pressure (Table 2). It has been determined that R_m_ is equal to 9.47 × 10^12^ m^−1^. This value is similar to the membranes resistance values presented in References [43,47], where UF membranes for purification of fermentation broths have been used (Appendix A, Table A1).

It has been found that during all performed UF experiments, the total filtration resistance to transport of permeating solvent gradually increased until the steady-state flux has been achieved. This phenomenon was caused by the concentration polarization and membrane fouling. Figure 6 illustrates the evolution of the total resistance, R_t_, during the UF process performed under TMP equal to 0.1 MPa and Q of 400 dm^3^/h. It has been noted that in the quasi-steady-state period of the process performed under the above-mentioned conditions, the R_t_ was equal to 2.38 × 10^13^ m^−1^.

It has been found that both transmembrane pressure and feed flow rate have a significant impact on the total resistance during the UF process of 1,3-PD fermentation broths (Figure 7). Results obtained in the present study clearly indicate that the total resistance increased as TMP increased and Q decreased. Indeed, the highest value of R_t_ (1.98 × 10^14^ m^−1^) has been reported during the process conducted under TMP equal to 0.4 MPa and Q of 200 dm^3^/h. The increasing of R_t_ with the increase of driving force can be caused by the fact that as TMP increases, more pollutants accumulate on the membrane surface, leading to the compression of the deposited layer. In turn, decreasing of Q led to an increase of R_t_, suggesting that the low value of the feed flow rate does not prevent the formation of the cake layer onto the membrane surface. These noteworthy results are similar to those presented by Yang et al. [85], where fouling analysis during the microfiltration process of cellulase fermentation broth has been presented. The authors have investigated the total resistance in function of transmembrane pressure and the feed cross-velocity. They have reported that the R_t_ significantly increased with an increase in the TMP from 0.05 to 0.02 MPa and, in turn, it decreased with an enhancement of the u from 2 to 4 m/s.

The contribution of different types of fouling resistance during the UF process has been investigated by using the resistance-in-series model. Figure 8 shows the contribution of membrane resistance and resistances of fouling (revisable and irreversible) in total resistance during the UF process of fermentation broths in function of TMP, under the feed flow rate equal to 200 dm^3^/h (a), 300 dm^3^/h (b) and 400 dm^3^/h (c). The percentage of membrane resistance was in the range between 4.8% (TMP = 0.4 MPa and Q = 200 dm^3^/h) and 27% (TMP = 0.1 MPa and Q = 400 dm^3^/h). The resistance of the membrane depends on the membrane characteristics and it is a constant value. Therefore, its contribution decreased with the increase in the overall resistance. In turn, it has been determined that the contribution of the reversible fouling resistance was between 28.0% (TMP = 0.1 MPa and Q = 400 dm^3^/h) and 45.9% (TMP = 0.2 MPa and Q = 200 dm^3^/h). It has been found that, generally, its percentage decreased with increasing the feed flow rate. Therefore, it can be indicated that a transition flow regime (Q of 400 dm^3^/h) reduces the contribution of reversible fouling. It confirms the results presented in Section 3.3., where it has been shown that increasing the turbulence near to the membrane led to a reduction in the mass and thickness of the cake layers formed on the membrane surface. This result is similar to findings obtained in Reference [94], where micro- and ultra-filtration processes of biological suspension with 5 g/dm^3^ of mixed liquor suspended solids have been investigated. The authors have shown that filtration resistances caused by concentration polarization and reversible fouling layer formed on the membranes surface significantly decreased with increasing the feed cross-flow-velocity, particularly from 0.1 to 2.0 m/s. In turn, in the present study, the contribution of the resistance caused by the irreversible fouling was in the range from 44% (TMP = 0.1 MPa and Q = 300 dm^3^/h) to 55% (TMP = 0.4 MPa and Q = 200 dm^3^/h). It has been found that the percentage of this resistance in the total resistance increased with the transmembrane pressure. It can be due to the fact that, as we mentioned above, the increasing of the process driving force led to the thickening or compression of the deposited layer onto the membrane surface. It has been noted that resistance caused by the irreversible fouling was dominant during the UF process performed under the following conditions: TMP = 0.4 MPa and Q = 400 dm^3^/h, TMP = 0.3 and 0.4 MPa, Q = 300 dm^3^/h, TMP = 0.3 and 0.4 MPa, Q = 200 dm^3^/h. It indicates that under the above-mentioned conditions, the effect of layer compression was stronger than that of the feed flow near the membrane surface.

## 4. Conclusions

The present study is part of the development process of 1,3-propanediol through fermentation of glycerol. The UF technique has been developed and demonstrated the complete removal of bacteria cells and macromolecular substances from fermentation broths containing 1,3-propanediol as the main product. Regardless of the process conditions applied, the ceramic membrane used in the present study provides the high-quality permeate that can be sequentially separated in NF and MD processes. Moreover, special attention has been paid to the impact of the operational parameters such as transmembrane pressure and feed flow rate on the permeate flux. Therefore, several UF experiments have been performed, under a wide range of TMP and Q values, from 0.1 to 0.4 MPa and 200 to 400 dm^3^/h, respectively. It has been found that among applied operational conditions, TMP equal to 0.4 MPa and Q of 400 dm^3^/h are the most favorable since they provide the highest permeate flux and the long-term performance stability. In turn, performing experiments on fermentation broths with a pH in the range of 5 to 10 allowed to indicate that the adjustment of the feed pH can cause a change in the interactions between feed and membrane, leading to the significant enhancement of ceramic membrane performance. The most favorable feed pH providing the highest process efficiency was equal to 9.4. It has been shown that by restoring the fouled membrane, its initial efficiency can be achieved by using 1% NaOH solution. The proposed method of membrane cleaning is effective, short and simple, which may have a positive effect on the economics of the UF process of 1,3-PD fermentation broths. Finally, the resistance-in-series model application allowed to determine that the contribution of the revisable fouling resistance in the total resistance decreases with increase in the feed flow rate and it is significantly reduced under a transition flow regime. In turn, the percentage of the resistance caused by the fouling irreversible increased with the increase of the process driving force. It is important to highlight that the membrane used shows the industrial potential of the presented application. Therefore, results obtained in the present study can assist in improving the cost-effectiveness of UF as a purification process of 1,3-PD fermentation broths from *Citrobacter freundii* and macromolecular substances.

## Figures and Tables

**Figure 1 membranes-10-00319-f001:**
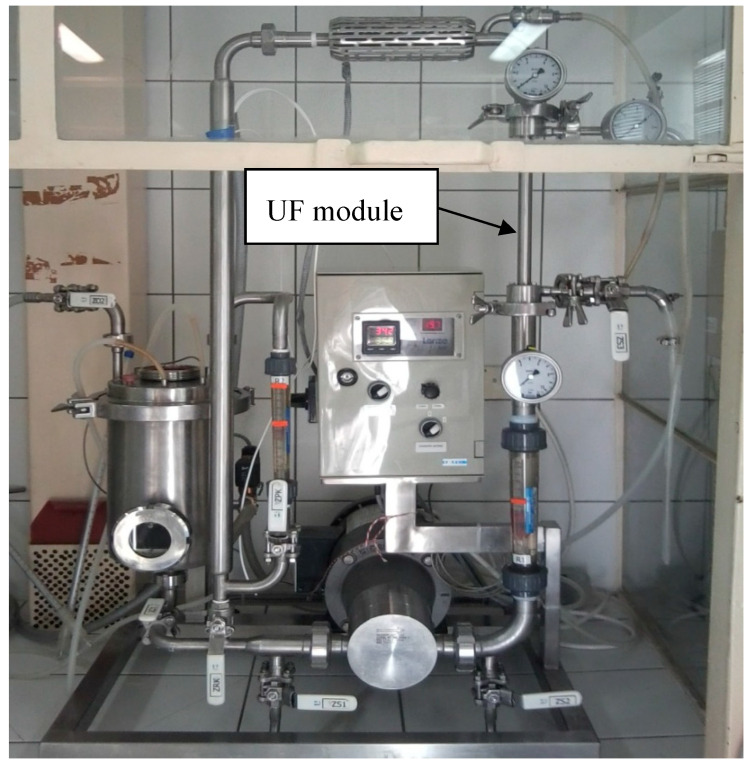
The photo of experimental ultrafiltration (UF) pilot-scale set-up.

**Figure 2 membranes-10-00319-f002:**
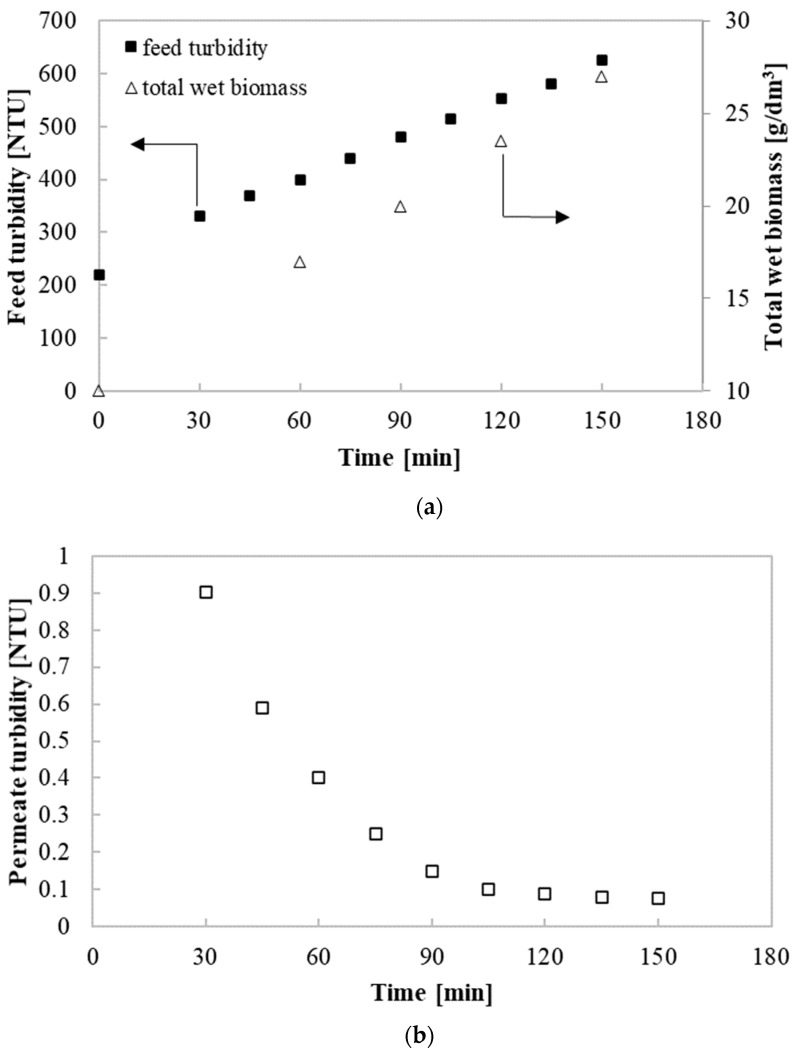
Changes of (**a**) feed turbidity and total wet biomass and (**b**) permeate turbidity during UF process.

**Figure 3 membranes-10-00319-f003:**
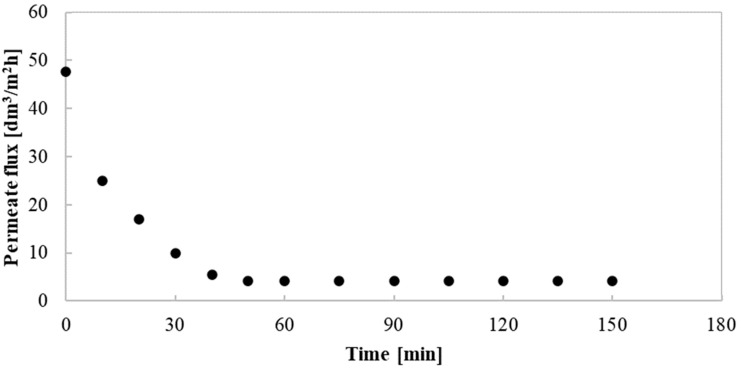
Changes of permeate flux during the UF process under TMP = 0.1 MPa and Q = 200 dm^3^/h.

**Figure 4 membranes-10-00319-f004:**
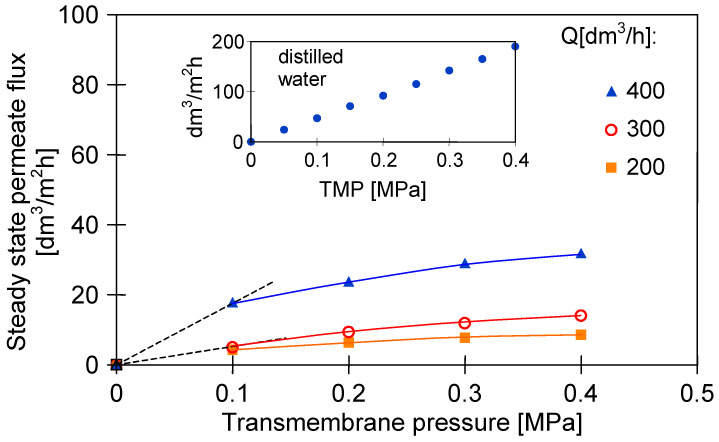
The impact of transmembrane pressure and feed flow rate on the steady-state permeate flux.

**Figure 5 membranes-10-00319-f005:**
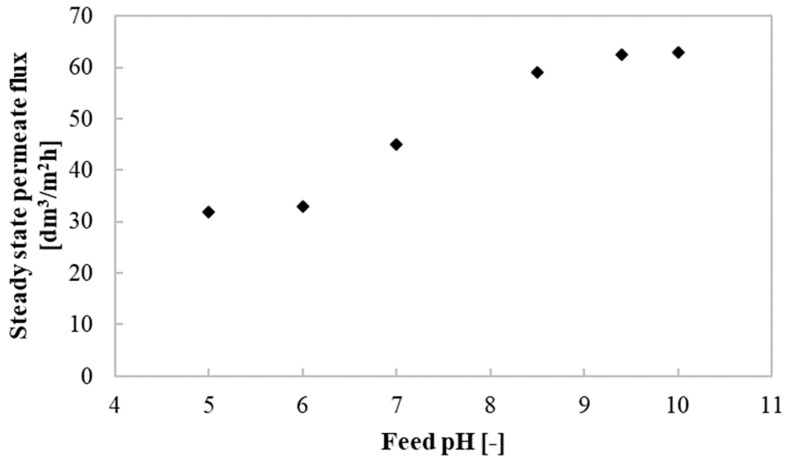
The impact of feed pH on the steady-state permeate flux. TMP = 0.4 MPa and Q = 400 dm^3^/h.

**Figure 6 membranes-10-00319-f006:**
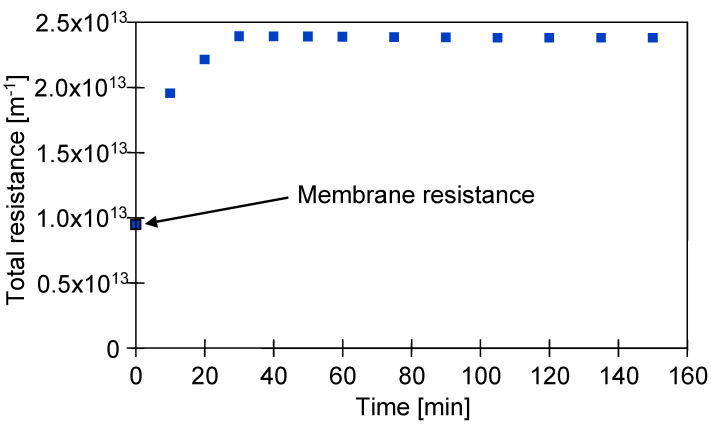
The evolution of the total resistance, R_t_, under TMP = 0.1 MPa and Q = 400 dm^3^/h.

**Figure 7 membranes-10-00319-f007:**
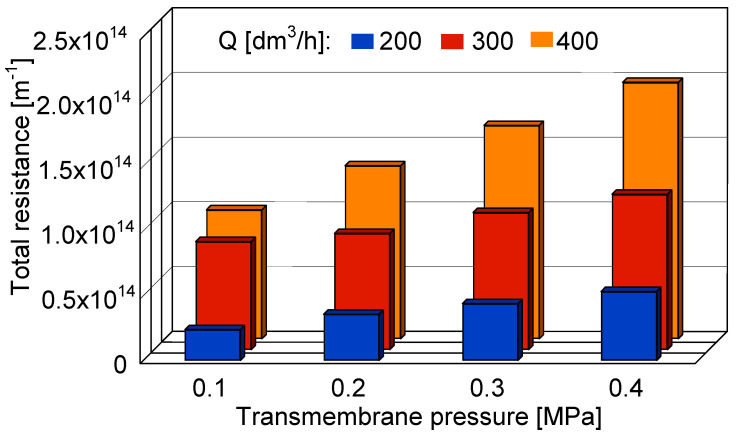
The impact of transmembrane pressure and feed flow rate on the total resistance.

**Figure 8 membranes-10-00319-f008:**
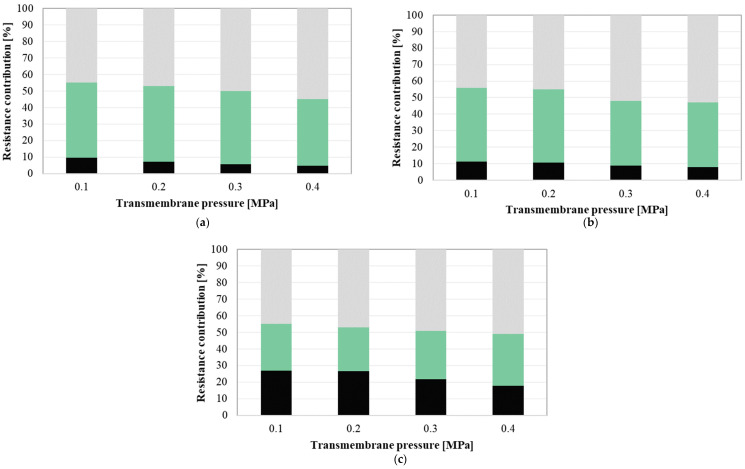
Resistances contribution as a function of transmembrane pressure. Q (dm^3^/h): (**a**) 200, (**b**) 300, (**c**) Q = 400 irreversible fouling, reversible fouling, membrane.

**Table 1 membranes-10-00319-t001:** Membrane specification.

Parameter	Unit	Value
Number of channels	(-)	1
Cut-off	(Da)	450
External diameter	(mm)	10
Channel diameter	(mm)	7
Filtration area per 0.215 m tube	(m^2^)	0.0047
Inflow area per tube	(mm^2^)	38.5
Support material	(-)	*α*-Al_2_O_3_
Support mean porosity, d_50_	(μm)	3
Open porosity	(%)	30–40
Membrane material	(-)	TiO_2_
Membrane mean porosity, d_50_	(nm)	0.9

**Table 2 membranes-10-00319-t002:** Description of parameters in Equation (4).

Symbol	Resistance	Determining Method	Formula	Comment
R_T_	total	experimental, based on steady-state flux for UF process	$R_{T}=\frac{\mathrm{TMP}}{\eta_{p}\cdot J}$	-
R_m_	clean membrane	experimental, based on water flux before UF experiments	$R_{m}=\frac{\mathrm{TMP}}{\eta_{w}\cdot J_{0}}$	*η*_w_—distilled water viscosity
R_irr_	irreversible fouling	calculated, based on water flux after membrane rinsing	$R_{\mathrm{irr}}=R_{T}-R_{m}-\frac{\mathrm{TMP}}{\eta_{w}\cdot J_{r}}$	J_r_—permeate flux for distilled water after membrane rinsing
R_rev_	reversible fouling	calculated, based on water flux after effective chemical cleaning of membrane	R_rev_ = R_T_ − R_m_ − R_irr_	-

**Table 3 membranes-10-00319-t003:** The compounds concentration in 1,3-propanodiol (1,3-PD) post-fermentation solutions with *Citrobacter freundii* bacteria.

Compound	1,3-PD	Citric Acid	Lactic Acid	Acetic Acid
Chemical Formula	HO(CH_2_)_3_OH	HOC(COOH)(CH_2_COOH)_2_	CH_3_CH(OH)COOH	CH_3_COOH
Concentration (g/L)	9.03–12.73	1.83–2.22	0.59–0.91	0.38–2.59

**Table 4 membranes-10-00319-t004:** The ions concentration in 1,3-PD post-fermentation solutions with *Citrobacter freundii* bacteria.

Ion	Cl^−^	NO_3_^−^	PO_4_^3−^	SO_4_^2−^	Na^+^	NH_4_^+^	K^+^	Ca^2+^	Mg^2+^
Concentration (g/L)	0.05–0.10	0.01–0.03	2.01–2.30	1.40–1.50	1.03–1.35	0.36–0.59	1.39–1.44	0.02–0.05	0.03–0.05

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
