# Peer review of "Clarification of 1,3-Propanediol Fermentation Broths by Using a Ceramic Fine UF Membrane"

_membranes, 2020, doi:10.3390/membranes10110319_

Round 1
Reviewer 1 Report
The introduction part of the UF mainly focuses on its application in post-fermentation solution. It would useful if authors could further summarize the state-of-art of the post-fermentation solution treatment process, the composition of some fermentation broths in general, including impurities, sizes, concentrations, etc. This would provide readers a clearer background why UF is needed, and how UF can fit in along the whole processing pathway.
Secondly, it would be useful if authors could compare different UF technologies, to explain why ceramic UF has been chosen for this study. Authors could also consider to highlight its main advantages and gaps over other UF membranes, highlight how these aspects are related to the post-fermentation solution separation process, permeate, retentate, and fouling, etc.
Author Response
Dear Editor and Reviewers,
We thank Reviewers for their interest in our work and for the valuable comments and constructive suggestions that greatly improve the manuscript. As indicated below, we have taken into considerations all general and specific comments provided by Reviewers and we have made changes and corrections accordingly to their indications.
Thank you for your time and effort.
Sincerely,
The authors
Review 1
- The introduction part of the UF mainly focuses on its application in post-fermentation solution. It would useful if authors could further summarize the state-of-art of the post-fermentation solution treatment process, the composition of some fermentation broths in general, including impurities, sizes, concentrations, etc. This would provide readers a clearer background why UF is needed, and how UF can fit in along the whole processing pathway.
According to the Reviewer’s comment, we have improved the Section 1. We have clearly indicated why UF is needed and how it can fit in along the whole processing pathway (lines 44-64). The state-of-art of the post-fermentation solutions treatment process is briefly discussed (lines 69-107).
- Secondly, it would be useful if authors could compare different UF technologies, to explain why ceramic UF has been chosen for this study. Authors could also consider to highlight its main advantages and gaps over other UF membranes, highlight how these aspects are related to the post-fermentation solution separation process, permeate, retentate, and fouling, etc.
According to the Reviewer's comment, we have indicated why ceramic UF membrane has been chosen, demonstrating its main advantages and gaps over other UF membranes (lines 97-104 and 114-126).
Reviewer 2 Report
This study used a ceramic fine ultrafiltration (UF) membrane for pre-treatment of glycerol fermentation broths containing 1,3-propanodiol (1,3-PD) as a main product. Special attention was paid to the impact of the operational parameters on the membrane performance. It was found that, TMP equal to 0.4 MPa and Q of 400 dm3/h ensure the highest permeate flux and its long-term stability. It is also reported the most favorable pH and the effective membrane cleaning method. Basically, the study may have certain value for real application. The manuscript is well orgainzed. Basically, this study should add to the literature. However, several concerns should be addressed before it can be considered for publication.
1). Introduction section: Too much general description on the UF process while less on the membrane fouling issue. I suggest to add one sentence in proper place: "Membrane fouling is a great concern for application of membrane technology (Water Res. 2020, 181,115932; J Colloid Interface Sci 2020, 565, 546-554; J. Membr Sci, 2020, 612: 118378)." The suggested refs. should be included. It is a serious concern.
2). Fig. 1 is too general and meaningless, and should be removed.
3). Why this membrane rather than others was used in this study?
4). Table 2: The authors tested the operation pressure of 0.1-0.4 MPa and concluded that 0.4 MPa is the best TMP. This is unlogical. Please explain it.
5). The major concern of this study is that, the authors observed some results, but didn't explain the causes at all. The causes of these behavors should be given.
6). The authors listed the Table A1, Is there any useful conclusion can be drawn?
Author Response
Dear Editor and Reviewers,
We thank Reviewers for their interest in our work and for the valuable comments and constructive suggestions that greatly improve the manuscript. As indicated below, we have taken into considerations all general and specific comments provided by Reviewers and we have made changes and corrections accordingly to their indications.
Thank you for your time and effort.
Sincerely,
The authors
Review 2
- Introduction section: Too much general description on the UF process while less on the membrane fouling issue. I suggest to add one sentence in proper place: "Membrane fouling is a great concern for application of membrane technology (Water Res. 2020, 181,115932; J Colloid Interface Sci 2020, 565, 546-554; J. Membr Sci, 2020, 612: 118378)." The suggested refs. should be included. It is a serious concern.
According to the Reviewer's comment, we have added this sentence and suggested references (lines 110-111).
- Fig. 1 is too general and meaningless, and should be removed.
We have removed the Figure 1.
- Why this membrane rather than others was used in this study?
The ultrafiltration membrane was selected based on a literature review which clearly shows that UF is an excellent technique for purification of post-fermentation solutions (lines 69-73). Regarding the membrane material, it has been well documented that ceramic membranes have many advantages over the other ones (lines 114-126). Moreover, performing a literature review allowed to indicate that the investigations of the purification of 1,3-PD fermentation broths by ceramic UF membranes are verry limited (97-99). Therefore, the present work allows to significantly expand the available knowledge in a given topic.
- Table 2: The authors tested the operation pressure of 0.1-0.4 MPa and concluded that 0.4 MPa is the best TMP. This is unlogical. Please explain it.
As a determinant of the best transmembrane pressure, we adopted the obtained permeate flux. Therefore, TMP equal to 0.4 MPa was the best, since it provides the highest permeate flux, equal to 31.78 dm3/m2h (for Q of 400 dm3/h) (lines 364-366).
- The major concern of this study is that, the authors observed some results, but didn't explain the causes at all. The causes of these behaviors should be given.
The explanations of the observed results are presented in our study. For instance:
- Section 3.2.: Changes of feed and permeate turbidity during UF process have been explained in lines 255-263,
- Section 3.3.: The impact of transmembrane pressure and feed flow rate on the steady state permeate flux has been explained in lines 327-329, 350-354, 360-363,
- Section 3.4.: The impact of feed pH on the steady state permeate flux has been explained in lines 399-404.
- Section 3.5.: The impact of transmembrane pressure and feed flow rate on the total resistance has been explained in lines 427-431
Resistances contribution as a function of transmembrane pressure has been explained in lines 450-453 and 460-466.
- The authors listed the Table A1, Is there any useful conclusion can be drawn?
This table has been listed in order to present the literature review on the UF process of fermentation broths. The following conclusions are presented in our paper:
(i) UF is the technique widely used for the present purpose (lines 69-73),
(ii) investigations of the purification of 1,3-PD fermentation broths by UF process are narrow (lines 97-99),
(iii) although several different types of UF membranes have been used to purify fermentation broths with other main products, the studies focused on the application of ceramic membranes are very limited (
lines 97-99),
(iii) although several different types of UF membranes have been used to purify fermentation broths with other main products, the studies focused on the application of ceramic membranes are very limited (lines 99-104).
Reviewer 3 Report
- The English needs to be polished. There are many grammatical and sentence structure errors. Please check throughout. For example:
- L10: examinated --> examined
- L41: have mentions
- L54: processes requires prior….
- Many more
- 1 is not needed. This is a general representation of UF and is a common knowledge. It would be better to show the schematic of the test set-up instead with details.
- Table 2: If only one line, it is not good to present in a table. Provide additional details, otherwise, include the information in the discussion. The same with Table 6.
- Can you please clarify what is the control of the experiments where you based the results for UF? Please present some comparative results using other membrane or process compared to UF to signify good potential of UF for the present purpose.
Author Response
Dear Editor and Reviewers,
We thank Reviewers for their interest in our work and for the valuable comments and constructive suggestions that greatly improve the manuscript. As indicated below, we have taken into considerations all general and specific comments provided by Reviewers and we have made changes and corrections accordingly to their indications.
Thank you for your time and effort.
Sincerely,
The authors
Review 3
- The English needs to be polished. There are many grammatical and sentence structure errors. Please check throughout. For example:
L10: examinated --> examined
L41: have mentions
L54: processes requires prior….
Many more
We have corrected grammatical and sentence structure errors in the paper indicated by Reviewer as well as others.
- 1 is not needed. This is a general representation of UF and is a common knowledge. It would be better to show the schematic of the test set-up instead with details.
We agree with Reviewer. We have removed the Figure 1 as it showed a common knowledge about the UF process. However, in our opinion, including the schematic of the test set-up is not necessary, since in the Section 2 we have presented the photo of the pilot installation used in our study.
Moreover, the installation diagram was included in the previous article in Membranes - there is Open Access, so everyone can see it without any problems.
- Table 2: If only one line, it is not good to present in a table. Provide additional details, otherwise, include the information in the discussion. The same with Table 6.
According to the Reviewer's comment, we have removed these tables and the information are included in the discussion.
- Can you please clarify what is the control of the experiments where you based the results for UF? Please present some comparative results using other membrane or process compared to UF to signify good potential of UF for the present purpose.
The control of the experiments where we based the results for UF are presented in the literature review in Section 1 (lines 69-96) as well as in: Section 3.2. (lines 281-285), Section 3.3. (lines 295-300, 329-331, 354-357), Section 3.5. (lines 453-457).
Comparative results using other technique compared to UF, signifying good potential of UF for the present purpose are presented in the Section 1 (lines 73-80).